# The Local Antimagic Chromatic Numbers of Some Join Graphs

**Xue Yang [1], Hong Bian [1,\*], Haizheng Yu [2] and Dandan Liu [1]**

[1] School of Mathematical Sciences, Xinjiang Normal University, Urumqi 830054, China; yx11092021@163.com (X.Y.); 633_ldd@sina.com (D.L.)

[2] College of Mathematics and System Sciences, Xinjiang University, Urumqi 830046, China; yuhaizheng@xju.edu.cn

\* Correspondence: bh1218@163.com

**Abstract:** Let $G = (V(G), E(G))$ be a connected graph with $n$ vertices and $m$ edges. A bijection $f : E(G) \to \{1, 2, \cdots, m\}$ is an edge labeling of $G$. For any vertex $x$ of $G$, we define $\omega(x) = \sum_{e \in E(x)} f(e)$ as the vertex label or weight of $x$, where $E(x)$ is the set of edges incident to $x$, and $f$ is called a local antimagic labeling of $G$, if $\omega(u) \neq \omega(v)$ for any two adjacent vertices $u, v \in V(G)$. It is clear that any local antimagic labelling of $G$ induces a proper vertex coloring of $G$ by assigning the vertex label $\omega(x)$ to any vertex $x$ of $G$. The local antimagic chromatic number of $G$, denoted by $\chi_{la}(G)$, is the minimum number of different vertex labels taken over all colorings induced by local antimagic labelings of $G$. In this paper, we present explicit local antimagic chromatic numbers of $F_n \vee \overline{K_2}$ and $F_n - v$, where $F_n$ is the friendship graph with $n$ triangles and $v$ is any vertex of $F_n$. Moreover, we explicitly construct an infinite class of connected graphs $G$ such that $\chi_{la}(G) = \chi_{la}(G \vee \overline{K_2})$, where $G \vee \overline{K_2}$ is the join graph of $G$ and the complement graph of complete graph $K_2$. This fact leads to a counterexample to a theorem of Arumugam et al. in 2017, and our result also provides a partial solution to Problem 3.19 in Lau et al. in 2021.

**Keywords:** local antimagic labeling; local antimagic chromatic number; join graph; friendship graph

## 1. Introduction

Throughout, we only consider undirected connected simple graphs. Let $G = (V(G), E(G))$ be a connected graph with $n$ vertices and $m$ edges. A bijection $f : E(G) \to \{1, 2, \cdots, m\}$ is an edge labeling of $G$. For any vertex $x$ of $G$, we define $\omega(x) = \sum_{e \in E(x)} f(e)$ as the vertex label or weight of $x$, where $E(x)$ is the set of edges incident to $x$, and $f$ is called an antimagic labeling of $G$, if $\omega(u) \neq \omega(v)$ for any two distinct vertices $u, v \in V(G)$. A graph $G$ is called antimagic if $G$ has an antimagic labeling.

The antimagic labeling of a graph was initially introduced by Hartsfield and Ringel [1] in 1990. They conjectured that every connected graph except $K_2$ admits such an antimagic labeling, which remains open till today.

Recently, based on the concept of antimagic labeling, Arumugam et al. [2] and Bensmail et al. [3] independently introduced the notation local antimagic labeling of graphs in 2017, which is weaker than antimagic labeling of graphs. Let $G = (V(G), E(G))$ be a connected graph of order $n$ and size $m$. A bijection $f : E(G) \to \{1, 2, \cdots, m\}$ is called a local antimagic labeling of $G$ if any two adjacent vertices $u$ and $v$ in $G$ satisfy $\omega(u) \neq \omega(v)$. It is clear that assigning $\omega(x)$ to $x$ for each $x \in V(G)$ naturally induced a proper vertex coloring of $G$, which is called a local antimagic vertex coloring of $G$. A graph $G$ is called local antimagic if $G$ has a local antimagic labeling. Haslegrave [4] showed that every connected graph with at least three vertices is local antimagic. The local antimagic chromatic number of $G$, denoted by $\chi_{la}(G)$, is the minimum number of different vertex labels taken over all colorings of $G$ induced by local antimagic labelings of $G$. If $f$ is a local antimagic labeling of $G$, the number of distinct induced vertex labels under $f$, denoted by $c(f)$, is called *the color number of $f$*.

A friendship graph, denoted by $F_n$, is a simple graph in which any two vertices have exactly one common neighbour, which consists of $n$ triangles with a common vertex. In [2], Arumugam et al. gave the exact value of the local antimagic chromatic numbers of special graphs, such as $P_n$, $C_n$, $F_n$, $K_{m,n}$, $K_{2,n}$, $W_n$, and $L(n)$, where $P_n$ and $C_n$ are path and cycle with $n$ vertices, respectively, $K_{m,n}$ is the complete bipartite graph ($m \equiv n \pmod 2$), $W_n$ is the wheel graph ($n \not\equiv 0 \pmod 4$), and $L(n)$ is the graph obtained by inserting a vertex to each edge of the star $S_n$. Ref. [5] was used in [2] to determine local antimagic chromatic numbers of complete bipartite graphs. When the graph is the wheel graph for $n \equiv 0 \pmod 4$ or the join graph $G \vee \overline{K_2}$ for $|V(G)| \geq 4$, where $\overline{K_2}$ is the complement graph of complete graph $K_2$, they also provided the lower and upper bounds of the local antimagic chromatic numbers of these graphs.

In 2018, Lau et al. [6] gave counterexamples to the lower bound of $\chi_{la}(G \vee \overline{K_2})$ that was obtained in [2]. Another counterexample was independently found by Shaebani [7]. A sharp lower bound of $\chi_{la}(G \vee \overline{K_n})$ and sufficient conditions for the given lower bound were obtained. Moreover, they gave affirmative solutions on Problem 3.3 of [2] and settled Theorem 2.15 of [2]. They also completely determined the local antimagic chromatic number of complete bipartite graphs.

In [8], Lau et al. provided several sufficient conditions for $\chi_{la}(H) \leq \chi_{la}(G)$, where $H$ is obtained from $G$ with a certain number of edge-deleted or -added operations. They then determined the exact values of the local antimagic chromatic numbers of many cycle-related join graphs.

In 2019, Lau et al. [9] gave the sharp lower bound of the local antimagic chromatic number of a graph with cut-vertices given by pendant edges and then solved Problem 3.3 in [2] affirmatively. In Section 2 of [9], Lau et al. gave sufficient conditions for the one-point union of cycles with $\chi_{la}(G) = 2$. In Section 3 of [9], they determined the exact values of the local antimagic chromatic numbers of many families of graphs with pendant edges. Finally, in Section 4, they obtained a few families of graphs with $\chi_{la}(G) = n$. This partially answered Problem 3.1 in [2].

Based on some known results, in this paper, we present the exact local antimagic chromatic numbers of $F_n \vee \overline{K_2}$ and $F_n - v$, where $v$ is any vertex of $F_n$. Moreover, we explicitly construct an infinite class of connected graphs $G$ such that $\chi_{la}(G) = \chi_{la}(G \vee \overline{K_2}) = 3$, where $G \vee \overline{K_2}$ is the join graph of $G$ and the complement graph of $K_2$. This fact leads to a counterexample to a theorem of [2], and our result also provides a partial solution to Problem 3.19 in [8].

## 2. Main Results

In [2], the authors gave the local antimagic chromatic number of the friendship graph as shown in the following lemma.

**Lemma 1** ([2])**.** *Let $F_n$ be a friendship graph, then we have $\chi_{la}(F_n) = 3$.*

For two vertex disjoint graphs $F_n$ and $\overline{K_2}$, let $F_n \vee \overline{K_2}$ denote the join graph obtained by joining every vertex of $F_n$ with every vertex of $\overline{K_2}$. In the proof of the local antimagic chromatic number of $F_n \vee \overline{K_2}$, we write $i \equiv t \pmod s$ $(0 \leq t < s)$ as $i \overset{s}{\equiv} t$ in the following formula. The following theorem gives an exact value of the local antimagic chromatic number of $F_n \vee \overline{K_2}$.

**Theorem 1.** *Let $H$ be the join graph $F_n \vee \overline{K_2}$, then we have $\chi_{la}(H) = 4$.*

**Proof.** Let $\{v, v_1, v_2, \cdots, v_{2n}\}$ be the vertex set of the friendship graph $F_n$, where $v$ is its central vertex, and let $x, y$ be the two vertices of $K_2$. It is clear that there are $7n + 2$ edges in $H$, namely, $\{v_i v_{i+1} : 1 \leq i \leq 2n \text{ and } i \equiv 1 \pmod 2\} \bigcup \{vv_i, xv_i, yv_i : 1 \leq i \leq 2n\} \bigcup \{xv, yv\}$. Since $K_4$ is an induced subgraph of $H$, we have $\chi_{la}(H) \geq \chi(H) \geq 4$. In order to prove

$\chi_{la}(H) = 4$, it suffices to provide a local antimagic labeling of $H$ that induces a local antimagic vertex coloring using exactly four colors.

We suppose that there is a local antimagic labeling $f : E(H) \rightarrow \{1, 2, 3, \cdots, 7n + 2\}$, such that $c(f) = 4$. It means that $\omega(v_1) = \omega(v_3) = \cdots = \omega(v_{2n-1})$, $\omega(v_2) = \omega(v_4) = \cdots = \omega(v_{2n})$, and $\omega(x) = \omega(y)$, which are distinct with $\omega(v)$. In this regard, we first assign $f(xv_i) = i$ or $4n + 1 - i$ and $f(yv_i) = 4n + 1 - i$ or $i$, for each $i \in \{1, 2, \cdots, 2n\}$, then determine the exact value of remaining edges of $H$. Let us consider the following four cases.

**Case 1.** $n \equiv 1 \pmod 4$

For $n = 1$, the graph $H = F_1 \vee \overline{K_2}$ admits a local antimagic labeling $f$ with $c(f) = 4$ as shown in Figure 1, which shows that $\chi_{la}(H) \leq 4$, and so $\chi_{la}(H) = 4$.

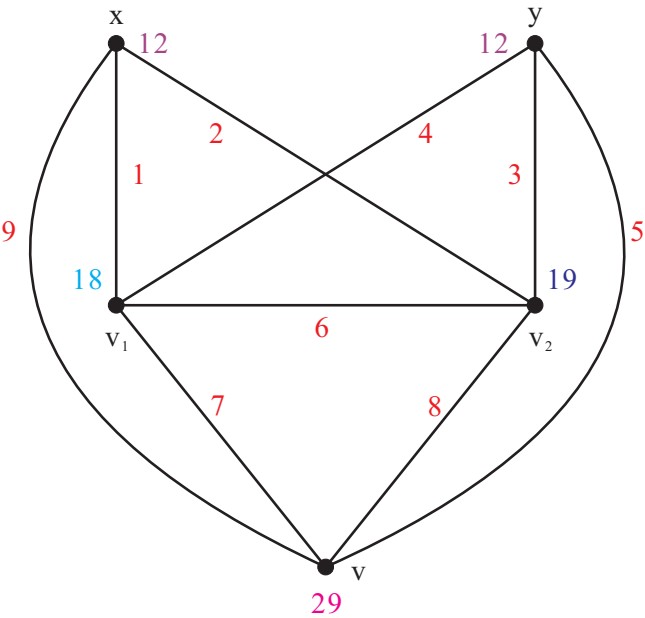

**Figure 1.** $F_1 \vee \overline{K_2}$.

For $n = 5$, we give the exact value of every edge label for the graph $H = F_5 \vee \overline{K_2}$ as shown in Figure 2.

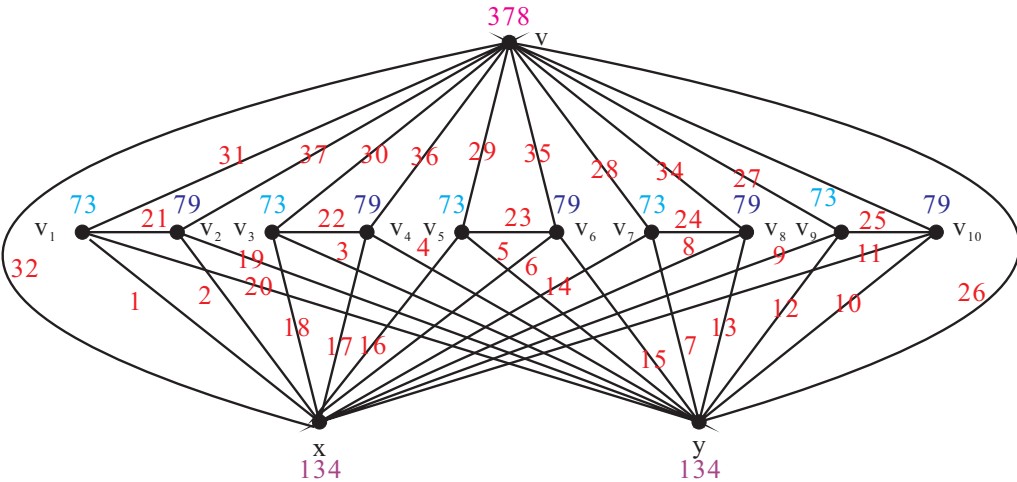

**Figure 2.** $F_5 \vee \overline{K_2}$.

It is obvious that

$$
\begin{aligned}
\omega(x) &= \omega(y) = 134, \\
\omega(v_1) &= \omega(v_3) = \omega(v_5) = \omega(v_7) = \omega(v_9) = 73, \\
\omega(v_2) &= \omega(v_4) = \omega(v_6) = \omega(v_8) = \omega(v_{10}) = 79, \\
\omega(v) &= 378.
\end{aligned}
$$

From the above labeling, $f$ is a local antimagic labeling of $H$ that induces a local antimagic vertex coloring using exactly four colors. It means that $\chi_{la}(H) \leq 4$, and so $\chi_{la}(H) = 4$.

For $n \geq 9$, define $f : E(H) \rightarrow \{1, 2, \cdots, 7n + 2\}$ in the following way:

Let $f(xv) = 6n + 2$, $f(yv) = 5n + 1$, and determine the values of $f(xv_i)$ and $f(yv_i)$ for each $i \in \{1, 2, 3, 4, 5, 2n - 4, 2n - 3, 2n - 2, 2n - 1, 2n\}$ as follows.

$$
f(xv_i) = \begin{cases} i, & \text{if } i \in \{1, 2, 2n - 4, 2n - 2, 2n - 1\}, \\ 4n + 1 - i, & \text{if } i \in \{3, 4, 5, 2n - 3, 2n\}. \end{cases}
$$

$$
f(yv_i) = \begin{cases} 4n + 1 - i, & \text{if } i \in \{1, 2, 2n - 4, 2n - 2, 2n - 1\}, \\ i, & \text{if } i \in \{3, 4, 5, 2n - 3, 2n\}. \end{cases}
$$

Then label the edges $xv_i$ and $yv_i$ for $6 \leq i \leq 2n - 5$, respectively.

$$
f(xv_i) = \begin{cases} i, & \text{if } i \overset{8}{\equiv} 6, \ i \overset{8}{\equiv} 0, \ i \overset{8}{\equiv} 1 \text{ or } i \overset{8}{\equiv} 2, \\ 4n + 1 - i, & \text{if } i \overset{8}{\equiv} 7, \ i \overset{8}{\equiv} 3, \ i \overset{8}{\equiv} 4 \text{ or } i \overset{8}{\equiv} 5. \end{cases}
$$

$$
f(yv_i) = \begin{cases} 4n + 1 - i, & \text{if } i \overset{8}{\equiv} 6, \ i \overset{8}{\equiv} 0, \ i \overset{8}{\equiv} 1 \text{ or } i \overset{8}{\equiv} 2, \\ i, & \text{if } i \overset{8}{\equiv} 7, \ i \overset{8}{\equiv} 3, \ i \overset{8}{\equiv} 4 \text{ or } i \overset{8}{\equiv} 5. \end{cases}
$$

Finally, we give the exact value of the remaining edges as follows.

$$
\begin{aligned}
f(v_i v_{i+1}) &= 4n + \tfrac{i+1}{2}, & i \overset{2}{\equiv} 1 \text{ and } 1 \leq i \leq 2n, \\
f(vv_i) &= 6n + 2 - \tfrac{i+1}{2}, & i \overset{2}{\equiv} 1, \\
f(vv_i) &= 7n + 3 - \tfrac{i}{2}, & i \overset{2}{\equiv} 0.
\end{aligned}
$$

Since $n \equiv 1 \pmod 4$ and $n \geq 9$, we have $2n \equiv 2 \pmod 8$, and so the number of vertices in $\{v_i | 6 \leq i \leq 2n - 5\}$ is divisible by 8.

If $\{i, i + 1, i + 2, \cdots, i + 7\} \subseteq \{6, 7, \cdots, 2n - 5\}$ and $i \equiv 6 \pmod 8$, then

$$
\sum_{j=i}^{i+7} f(xv_j) = 16n - 6, \quad \sum_{j=i}^{i+7} f(yv_j) = 16n + 14.
$$

Accordingly, we have

$$
\sum_{i=6}^{2n-5} f(xv_i) = 4n^2 - \frac{43n}{2} + \frac{15}{2},
$$

$$
\sum_{i=6}^{2n-5} f(yv_i) = 4n^2 - \frac{33n}{2} - \frac{35}{2}.
$$

Since

$$
\sum_{i=1}^{5} f(xv_i) = 12n - 6, \quad \sum_{i=2n-4}^{2n} f(xv_i) = 10n - 2,
$$

$$\sum_{i=1}^{5} f(yv_i) = 8n + 11, \qquad \sum_{i=2n-4}^{2n} f(xv_i) = 10n + 7.$$

It is clear that $f$ is a local antimagic labeling of $H$ and

$$\omega(x) = \omega(y) = 4n^2 + \tfrac{13n}{2} + \tfrac{3}{2},$$
$$\omega(v_i) = 14n + 3, \quad i \overset{2}{\equiv} 1,$$
$$\omega(v_i) = 15n + 4, \quad i \overset{2}{\equiv} 0,$$
$$\omega(v) = 12n^2 + 15n + 3.$$

Hence, $\chi_{la}(H) \leq 4$. The local antimagic labeling of the graph $F_9 \vee \overline{K_2}$ is shown in Figure 3.

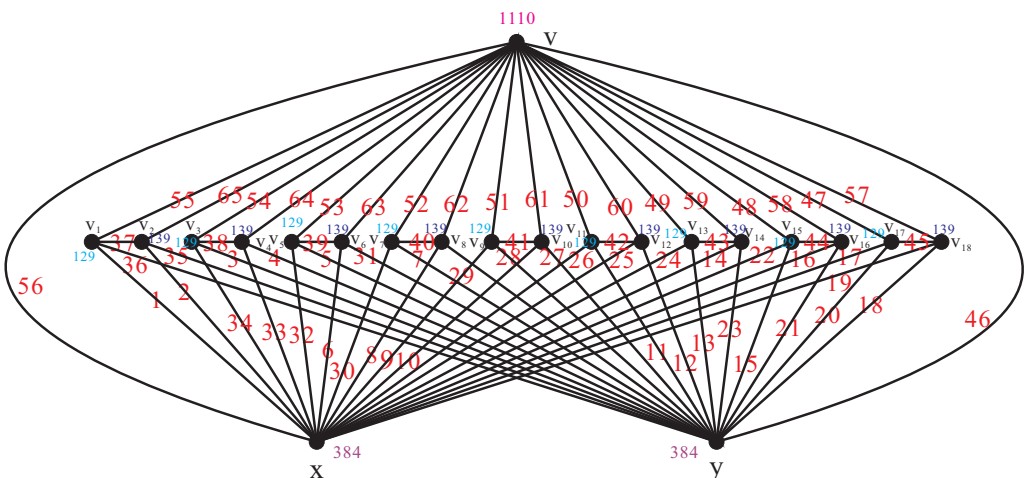

**Figure 3.** $F_9 \vee \overline{K_2}$.

**Case 2.** $n \equiv 3 \pmod 4$

For $n = 3$ as shown in Figure 4, we obtain a local antimagic labeling of $F_3 \vee \overline{K_2}$ with $c(f) = 4$.

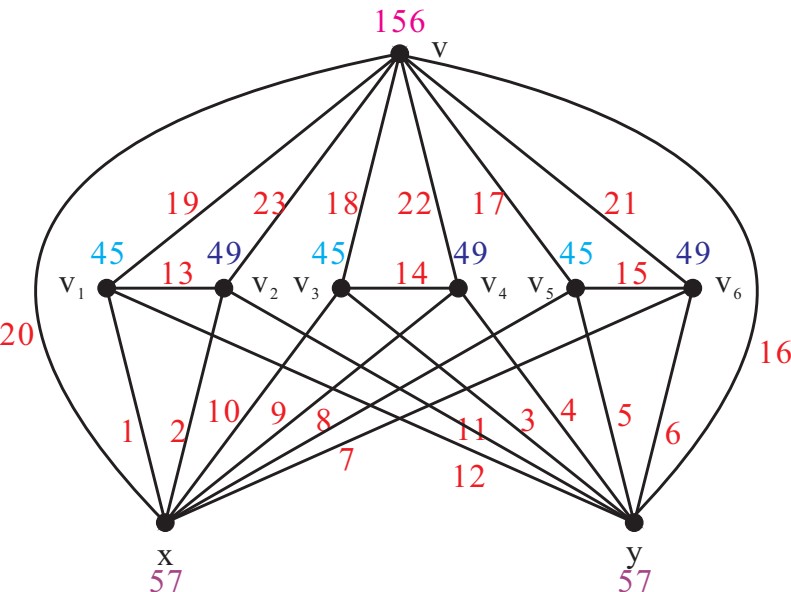

**Figure 4.** $F_3 \vee \overline{K_2}$.

For $n \geq 7$, define $f : E(H) \rightarrow \{1, 2, \cdots, 7n + 2\}$ by the following

$$f(xv) = 6n + 2, \ f(yv) = 5n + 1.$$

Firstly, we set the following assignments of $xv_i$ and $yv_i$ for some special $i$, respectively.

$$f(xv_i) = \begin{cases} i, & \text{if } i \in \{1, 2\}, \\ 4n + 1 - i, & \text{if } i \in \{3, 4, 5, 2n\}. \end{cases}$$

$$f(yv_i) = \begin{cases} 4n + 1 - i, & \text{if } i \in \{1, 2\}, \\ i, & \text{if } i \in \{3, 4, 5, 2n\}. \end{cases}$$

Secondly, considering the following assignments of the edges $xv_i$ and $yv_i$ for $6 \leq i \leq 2n - 1$,

$$f(xv_i) = \begin{cases} i, & \text{if } i \overset{8}{\equiv} 6, \ i \overset{8}{\equiv} 0, \ i \overset{8}{\equiv} 1 \text{ or } i \overset{8}{\equiv} 2, \\ 4n + 1 - i, & \text{if } i \overset{8}{\equiv} 7, \ i \overset{8}{\equiv} 3, \ i \overset{8}{\equiv} 4 \text{ or } i \overset{8}{\equiv} 5. \end{cases}$$

$$f(yv_i) = \begin{cases} 4n + 1 - i, & \text{if } i \overset{8}{\equiv} 6, \ i \overset{8}{\equiv} 0, \ i \overset{8}{\equiv} 1 \text{ or } i \overset{8}{\equiv} 2, \\ i, & \text{if } i \overset{8}{\equiv} 7, \ i \overset{8}{\equiv} 3, \ i \overset{8}{\equiv} 4 \text{ or } i \overset{8}{\equiv} 5. \end{cases}$$

Finally, label the remaining edges as follows

$$f(v_i v_{i+1}) = 4n + \tfrac{i+1}{2}, \quad i \overset{2}{\equiv} 1 \text{ and } 1 \leq i \leq 2n,$$
$$f(vv_i) = 6n + 2 - \tfrac{i+1}{2}, \quad i \overset{2}{\equiv} 1,$$
$$f(vv_i) = 7n + 3 - \tfrac{i}{2}, \quad i \overset{2}{\equiv} 0.$$

Because $n \equiv 3 \pmod 4$ and $n \geq 7$, we have $2n \equiv 6 \pmod 8$, and so the number of vertices in $\{v_i | 6 \leq i \leq 2n - 1\}$ is divisible by 8.

If $\{i, i+1, i+2, \cdots, i+7\} \subseteq \{6, 7, \cdots, 2n - 1\}$ and $i \equiv 6 \pmod 8$, then

$$\sum_{j=i}^{i+7} f(xv_j) = 16n - 6, \quad \sum_{j=i}^{i+7} f(yv_j) = 16n + 14.$$

We can obtain that
$$\sum_{i=6}^{2n-1} f(xv_i) = 4n^2 - \frac{27n}{2} + \frac{9}{2},$$

$$\sum_{i=6}^{2n-1} f(yv_i) = 4n^2 - \frac{17n}{2} - \frac{21}{2}.$$

Since
$$\sum_{i=1}^{5} f(xv_i) = 12n - 6, \quad \sum_{i=1}^{5} f(yv_i) = 8n + 11,$$

$$f(xv_{2n}) = 2n + 1, \quad f(yv_i) = 2n.$$

For the vertex weights we have

$$\omega(x) = \omega(y) = 4n^2 + \tfrac{13n}{2} + \tfrac{3}{2},$$
$$\omega(v_i) = 14n + 3, \quad i \overset{2}{\equiv} 1,$$
$$\omega(v_i) = 15n + 4, \quad i \overset{2}{\equiv} 0,$$
$$\omega(v) = 12n^2 + 15n + 3.$$

Hence, we can obtain that $\chi_{la}(H) = 4$. For $n = 7$, the exact values of each edge label of the graph $F_7 \vee \overline{K_2}$ are given in Figure 5.

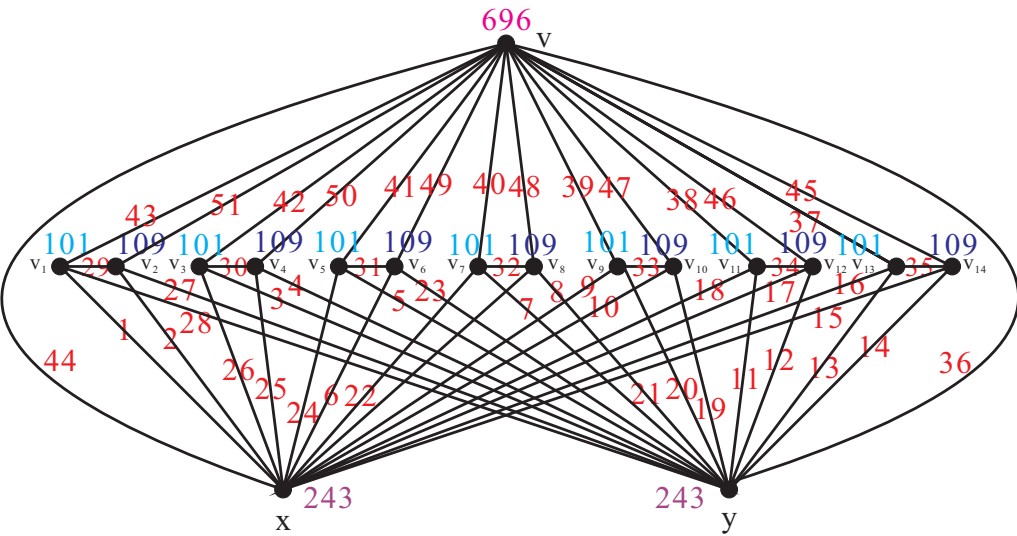

**Figure 5.** $F_7 \vee \overline{K_2}$.

**Case 3.** $n \equiv 2 \pmod 4$

In this case, we consider $n \equiv 2 \pmod 8$ and $n \equiv 6 \pmod 8$, respectively.

Subcase 3.1. $n \equiv 2 \pmod 8$

For $n = 2$, there is a local antimagic labeling of the graph $H = F_2 \vee \overline{K_2}$ in Figure 6. Hence, we have $\chi_{la}(H) = 4$.

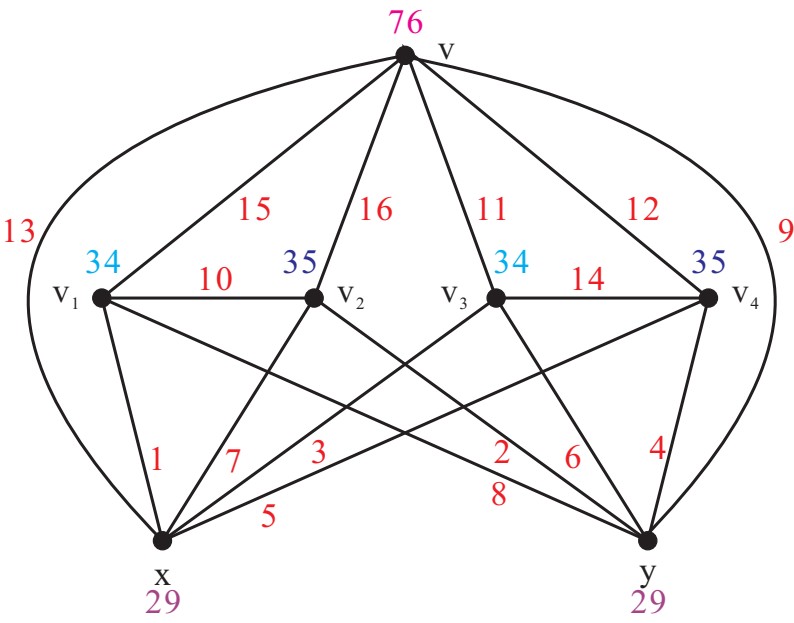

**Figure 6.** $F_2 \vee \overline{K_2}$.

For $n \geq 10$, define the edge labeling $f : E(H) \rightarrow \{1, 2, \cdots, 7n + 2\}$ as follows:

$$f(xv) = \frac{11n}{2} + 2, \quad f(yv) = 4n + 1.$$

Assume that $n = 8k + 2$, $k = 1, 2, \cdots$, then we give the following exact values of $f(xv_i)$ and $f(yv_i)$ for $1 \leq i \leq 2n$.

$$f(xv_i) = \begin{cases} 4n + 1 - i, & \text{if } 1 \le i \le 2k \text{ and } i \overset{2}{\equiv} 1, \\ i, & \text{if } 1 \le i \le 2k \text{ and } i \overset{2}{\equiv} 0, \\ i, & \text{if } 2k + 1 \le i \le 2n \text{ and } i \overset{2}{\equiv} 1, \\ 4n + 1 - i, & \text{if } 2k + 1 \le i \le 2n \text{ and } i \overset{2}{\equiv} 0. \end{cases}$$

$$f(yv_i) = \begin{cases} i, & \text{if } 1 \le i \le 2k \text{ and } i \overset{2}{\equiv} 1, \\ 4n + 1 - i, & \text{if } 1 \le i \le 2k \text{ and } i \overset{2}{\equiv} 0, \\ 4n + 1 - i, & \text{if } 2k + 1 \le i \le 2n \text{ and } i \overset{2}{\equiv} 1, \\ i, & \text{if } 2k + 1 \le i \le 2n \text{ and } i \overset{2}{\equiv} 0. \end{cases}$$

Then label the remaining edges as follows:

$$f(v_i v_{i+1}) = \begin{cases} 4n + 1 + \frac{i+1}{2}, & \text{if } 1 \le i \le n \text{ and } i \overset{2}{\equiv} 1, \\ 5n + 2 + \frac{i+1}{2}, & \text{if } n + 1 \le i \le 2n \text{ and } i \overset{2}{\equiv} 1. \end{cases}$$

$$f(vv_i) = \begin{cases} \frac{13n+4}{2} - \frac{i-1}{2}, & \text{if } 1 \le i \le n \text{ and } i \overset{2}{\equiv} 1, \\ 7n + 3 - \frac{i}{2}, & \text{if } 1 \le i \le n \text{ and } i \overset{2}{\equiv} 0, \\ \frac{11n+2}{2} - \frac{i-1}{2}, & \text{if } n + 1 \le i \le 2n \text{ and } i \overset{2}{\equiv} 1, \\ 6n + 2 - \frac{i}{2}, & \text{if } n + 1 \le i \le 2n \text{ and } i \overset{2}{\equiv} 0. \end{cases}$$

It is clear that $f$ is a local antimagic labeling of $H$, and we have

$$\omega(x) = \omega(y) = 4n^2 + \frac{23n}{4} + \frac{3}{2},$$
$$\omega(v_i) = \frac{29n}{2} + 5, \quad i \overset{2}{\equiv} 1,$$
$$\omega(v_i) = 15n + 5, \quad i \overset{2}{\equiv} 0,$$
$$\omega(v) = \frac{23n^2}{2} + \frac{27n}{2} + 3.$$

So, we have $\chi_{la}(F_n \vee \overline{K_2}) = 4$ for $n \overset{8}{\equiv} 2$. The local antimagic labeling of the graph $F_{10} \vee \overline{K_2}$ is shown in Figure 7.

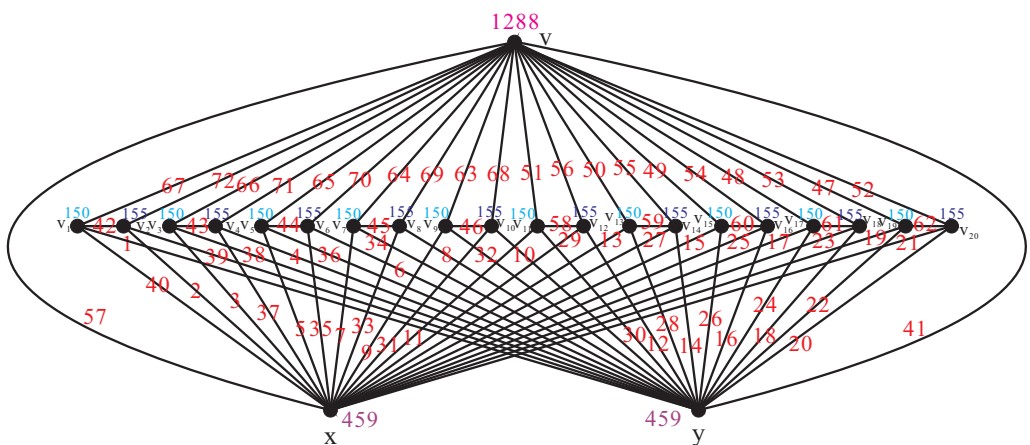

**Figure 7.** $F_{10} \vee \overline{K_2}$.

Subcase 3.2. $n \equiv 6 \pmod 8$

For $n \ge 6$, label the edges of $H$ by the labeling $f : E(H) \to \{1, 2, \cdots, 7n + 2\}$ such that

$$f(xv) = \frac{11n}{2} + 2, \quad f(yv) = 4n + 1.$$

Assume $n = 8k - 2$, $k = 1, 2, \cdots$, then we label $f(xv_i)$ and $f(yv_i)$ for each $i$ such that $1 \le i \le 2n - 1$.

$$f(xv_i) = \begin{cases} 4n + 1 - i, & \text{if } 1 \le i \le 2k \text{ and } i \overset{2}{\equiv} 1, \\ i, & \text{if } 1 \le i \le 2k \text{ and } i \overset{2}{\equiv} 0, \\ i, & \text{if } 2k + 1 \le i \le 2n - 1 \text{ and } i \overset{2}{\equiv} 1, \\ 4n + 1 - i, & \text{if } 2k + 1 \le i \le 2n - 1 \text{ and } i \overset{2}{\equiv} 0. \end{cases}$$

$$f(yv_i) = \begin{cases} i, & \text{if } 1 \le i \le 2k \text{ and } i \overset{2}{\equiv} 1, \\ 4n + 1 - i, & \text{if } 1 \le i \le 2k \text{ and } i \overset{2}{\equiv} 0, \\ 4n + 1 - i, & \text{if } 2k + 1 \le i \le 2n - 1 \text{ and } i \overset{2}{\equiv} 1, \\ i, & \text{if } 2k + 1 \le i \le 2n - 1 \text{ and } i \overset{2}{\equiv} 0. \end{cases}$$

For the last vertex $v_{2n}$,

$$f(xv_{2n}) = 2n, \ f(yv_{2n}) = 2n + 1.$$

Now, determine the exact value of $f(vv_i)$ for each $i$ such that $1 \le i \le 2n$.

$$f(vv_i) = \begin{cases} \frac{13n+4}{2} - \frac{i-1}{2}, & \text{if } 1 \le i \le n \text{ and } i \overset{2}{\equiv} 1, \\ 7n + 3 - \frac{i}{2}, & \text{if } 1 \le i \le n \text{ and } i \overset{2}{\equiv} 0, \\ \frac{11n+2}{2} - \frac{i-1}{2}, & \text{if } n + 1 \le i \le 2n \text{ and } i \overset{2}{\equiv} 1, \\ 6n + 2 - \frac{i}{2}, & \text{if } n + 1 \le i \le 2n \text{ and } i \overset{2}{\equiv} 0. \end{cases}$$

When $i$ is odd for $1 \le i \le 2n$, we can label $f(v_i v_{i+1})$ as follows.

$$f(v_i v_{i+1}) = \begin{cases} 4n + 1 + \frac{i+1}{2}, & \text{if } 1 \le i \le n \text{ and } i \overset{2}{\equiv} 1, \\ 5n + 2 + \frac{i+1}{2}, & \text{if } n + 1 \le i \le 2n \text{ and } i \overset{2}{\equiv} 1. \end{cases}$$

For the vertex weights under the labeling $f$, we have

$$\begin{aligned} \omega(x) = \omega(y) &= 4n^2 + \frac{23n}{4} + \frac{3}{2}, \\ \omega(v_i) &= \frac{29n}{2} + 5, \ i \overset{2}{\equiv} 1, \\ \omega(v_i) &= 15n + 5, \ i \overset{2}{\equiv} 0, \\ \omega(v) &= \frac{23n^2}{2} + \frac{27n}{2} + 3. \end{aligned}$$

This implies that $\chi_{la}(H) = 4$. For $n = 6$, we obtain the local antimagic labeling of the graph $F_6 \vee \overline{K_2}$ under $f$ as shown in Figure 8.

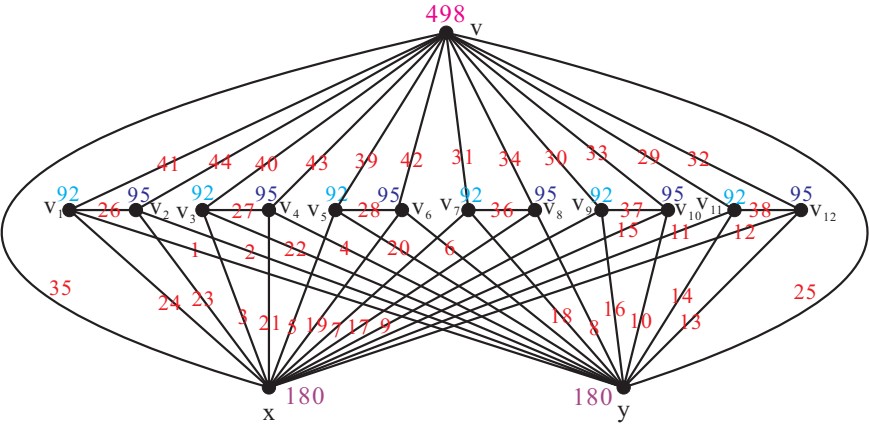

**Figure 8.** $F_6 \vee \overline{K_2}$.

**Case 4.** $n \equiv 0 \pmod 4$

We define $f : E(H) \rightarrow \{1, 2, \cdots, 7n+2\}$ as follows:

$$f(xv) = 4n+3, \quad f(yv) = 4n+1.$$

The following labeling has the desired properties:

$$f(xv_i) = \begin{cases} 4n+1-i, & \text{if } i \stackrel{4}{\equiv} 1 \text{ or } i \stackrel{4}{\equiv} 0, \text{ and } i \neq 2n, \\ i, & \text{if } i \stackrel{4}{\equiv} 3 \text{ or } i \stackrel{4}{\equiv} 2, \text{ or } i = 2n. \end{cases}$$

$$f(yv_i) = \begin{cases} i, & \text{if } i \stackrel{4}{\equiv} 1 \text{ or } i \stackrel{4}{\equiv} 0, \text{ and } i \neq 2n, \\ 4n+1-i, & \text{if } i \stackrel{4}{\equiv} 3 \text{ or } i \stackrel{4}{\equiv} 2, \text{ or } i = 2n. \end{cases}$$

$$f(v_i v_{i+1}) = \begin{cases} 5n+2+i, & \text{if } 1 \leq i \leq n+1 \text{ and } i \stackrel{2}{\equiv} 1, \\ 7n+3-i, & \text{if } n+3 \leq i \leq 2n \text{ and } i \stackrel{2}{\equiv} 1. \end{cases}$$

$$f(vv_i) = \begin{cases} 5n+3-i, & \text{if } 1 \leq i \leq n+2 \text{ and } i \stackrel{2}{\equiv} 1, \\ 7n+4-i, & \text{if } 1 \leq i \leq n+2 \text{ and } i \stackrel{2}{\equiv} 0, \\ 3n+2+i, & \text{if } n+3 \leq i \leq 2n \text{ and } i \stackrel{2}{\equiv} 1, \\ 5n+1+i, & \text{if } n+3 \leq i \leq 2n \text{ and } i \stackrel{2}{\equiv} 0. \end{cases}$$

For the vertex weights under the labeling $f$, we have

$$\omega(x) = \omega(y) = 4n^2 + 5n + 2,$$
$$\omega(v_i) = 14n + 6, \quad i \stackrel{2}{\equiv} 1,$$
$$\omega(v_i) = 16n + 6, \quad i \stackrel{2}{\equiv} 0.$$
$$\omega(v) = 11n^2 + 13n + 2.$$

The above arguments indicate that $f$ is a local antimagic labeling of $H$ with four colors, and so $\chi_{la}(H) = 4$. The exact values of each edge label of the graph $F_4 \vee \overline{K}_2$ are given in Figure 9. The proof is completed. $\square$

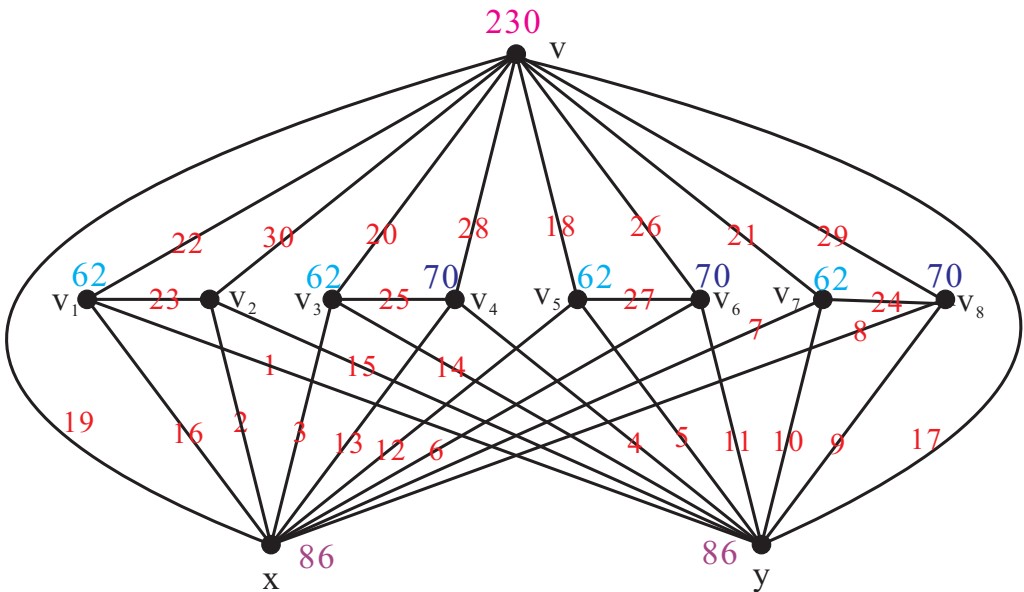

**Figure 9.** $F_4 \vee \overline{K}_2$.

Let $H = F_n - v$ be a graph obtained from the friendship graph $F_n (n \geq 2)$ by deleting any vertex $v$ of $F_n$. If the deleted vertex is its central vertex, then $H$ does not have a local antimagic labeling. Thus, we only consider that the deleted vertex is a vertex with degree 2.

**Theorem 2.** *Let $H$ be the graph $F_n - v$, where $v$ is any vertex of $F_n (n \geq 2)$ with degree 2, then we have $\chi_{la}(H) = 3$.*

**Proof.** Let $V(F_n) = \{u_i : 1 \leq i \leq n\} \cup \{v_i : 1 \leq i \leq n\} \cup \{x\}$ and $E(F_n) = \{u_i v_i : 1 \leq i \leq n\} \cup \{xu_i : 1 \leq i \leq n\} \cup \{xv_i : 1 \leq i \leq n\}$. Without loss of generality, we assume that the deleted vertex is $v_n \in V(F_n)$, then define $h : E(H) \rightarrow \{1, 2, \cdots, 3n-2\}$ by

$$
\begin{aligned}
h(u_i v_i) &= i, & 1 \leq i \leq n-1, \\
h(xu_i) &= 3n-2-i, & 1 \leq i \leq n-1, \\
h(xv_i) &= 2n-1-i, & 1 \leq i \leq n-1, \\
h(xu_n) &= 3n-2.
\end{aligned}
$$

Clearly, $h$ is a local antimagic labeling of $H$ and we have

$$
\begin{aligned}
\omega(v_i) &= 2n-1, \text{ where } 1 \leq i \leq n-1, \\
\omega(u_i) &= 3n-2, \text{ where } 1 \leq i \leq n, \\
\omega(x) &= 4n^2 - 4n + 1.
\end{aligned}
$$

Thus, $\chi_{la}(H) \leq 3$. Since $\chi_{la}(H) \geq \chi(H) = 3$; it follows that $\chi_{la}(H) = 3$. $\square$

Theorem 2.16 of [2] asserts that if a graph $G$ has at least four vertices, then $\chi_{la}(G) + 1 = \chi_{la}(G \vee \overline{K_2})$, when $G$ is of even order $n$. In this section, we explicitly construct an infinite class of connected graphs $G$ such that $\chi_{la}(G) = 3$ and $\chi_{la}(G \vee \overline{K_2}) = 3$. Our procedure is to consider path $P_n$ that satisfies $\chi_{la}(P_n) = 3$ for each positive integer $n \geq 3$. We show that if $n$ is even, then $\chi_{la}(P_n \vee \overline{K_2}) = 3$. Our result provides partial solution to Problem 3.19 in [8].

**Theorem 3.** *If $P_n$ is a path of order $n$, then we have $\chi_{la}(P_n \vee \overline{K_2}) = 3$ for even $n$.*

**Proof.** The lower bound of the local antimagic chromatic number of the join graph $P_n \vee \overline{K_2}$ even for $n$ is clearly obtained. We have $\chi_{la}(P_n \vee \overline{K_2}) \geq \chi(P_n \vee \overline{K_2}) = 3$ since $K_3$ is a induced subgraph of $P_n \vee \overline{K_2}$. We show that the upper bound of the chromatic number $\chi_{la}(P_n \vee \overline{K_2})$ is attainable.

Let $\{u_i : 1 \leq i \leq n\}$ and $\{x, y\}$ be the vertex set of the path $P_n$ and the complement graph of $K_2$, respectively. Then $E(P_n \vee \overline{K_2}) = \{xu_i, yu_i : 1 \leq i \leq n\} \cup \{u_i u_{i+1} : 1 \leq i \leq n-1\}$, and $|E(P_n \vee \overline{K_2})| = 3n-1$.

Label the edges $u_i u_{i+1}$ as follows:

$$
f(u_i u_{i+1}) = \begin{cases} n - \frac{i+1}{2}, & \text{if } i \text{ is odd}, \\ \frac{i}{2}, & \text{if } i \text{ is even}. \end{cases}
$$

Then, label the edges $xu_i$ as follows:

$$
f(xu_i) = \begin{cases} n + \frac{i-1}{2}, & \text{if } i \text{ is odd}, \\ 3n - \frac{i+2}{2}, & \text{if } i \text{ is even}, i \neq n, \\ 3n-1, & i = n. \end{cases}
$$

Finally, label the edges $yu_i$ as follows:

$$
f(yu_i) = \begin{cases} \frac{5n}{2} - \frac{i+1}{2}, & \text{if } i \text{ is odd}, \\ \frac{3n}{2} + \frac{i-2}{2}, & \text{if } i \text{ is even}. \end{cases}
$$

We can conclude that

$$\omega(u_i) = \frac{9n}{2} - 2, \qquad \text{if } i \text{ is odd;}$$
$$\omega(u_i) = \frac{11n}{2} - 2, \qquad \text{if } i \text{ is even;}$$
$$\omega(x) = \omega(y) = 2n^2 - \frac{n}{2}.$$

Therefore, $f$ is a local antimagic labeling of $P_n \vee \overline{K_2}$ that induces a local antimagic vertex coloring using exactly three colors. The local antimagic labeling of the graph $P_6 \vee \overline{K_2}$ as an example is shown in Figure 10. $\square$

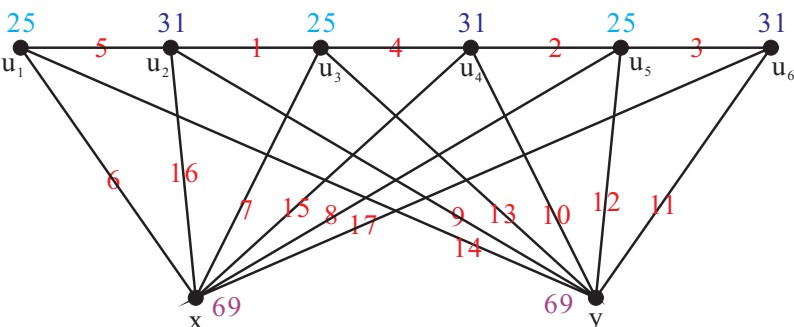

**Figure 10.** $P_6 \vee \overline{K_2}$.

### 3. Conclusions and Scope

In this paper, we obtain the exact values of the local antimagic chromatic number of the join graphs $F_n \vee \overline{K_2}$, $P_n \vee \overline{K_2}$ and the graph $F_n - v$. Hence, the following problem arises naturally.

**Problem 1.** *Find the local antimagic chromatic number of the cartesian product of simple graphs $G$ and $H$.*

**Problem 2.** *Find the local antimagic chromatic number of other operations of graphs.*

**Problem 3.** *Characterize the class of a graph $G$ for which $\chi_{la}(G \vee \overline{K_2}) = \chi_{la}(G)$.*

**Author Contributions:** Conceptualization, X.Y. and H.B.; methodology, X.Y.; software, H.Y.; validation, X.Y., H.B. and H.Y.; formal analysis, D.L.; investigation, X.Y.; resources, H.B.; data curation, H.Y.; writing—original draft preparation, X.Y.; writing—review and editing, H.B.; visualization, D.L.; supervision, H.B.; project administration, H.B.; funding acquisition, H.B. All authors have read and agreed to the published version of the manuscript.

**Funding:** The research and publication of our article was funded by the National Natural Science Foundation of China (11761070, 61662079) and 2021 Xinjiang Uygur Autonomous Region National Natural Science Foundation Joint Research Fund (2021D01C078), 2020 Special Foundation for First-class Specialty of Applied Mathematics Xinjiang Normal University. H. Bian was supported by the National Natural Science Foundation of China (11761070) and 2020 Special Foundation for First-class Specialty of Applied Mathematics Xinjiang Normal University. H. Z. Yu was supported by the National Natural Science Foundation of China (61662079) and 2021 Xinjiang Uygur Autonomous Region National Natural Science Foundation Joint Research Fund (2021D01C078).

**Acknowledgments:** The authors thank the anonymous referees for their helpful suggestions to improve the exposition.

**Conflicts of Interest:** The authors declare that there is no conflict of interest regarding the publication of this paper.

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
