# Peer review of "The Local Antimagic Chromatic Numbers of Some Join Graphs"

_mca, doi:10.3390/mca26040080_

Round 1
Reviewer 1 Report
This paper studies the local antimagic chromatic number, introduced by Arumugam et al., of some specific graphs. There has been quite a lot of work on this parament, by various authors, since its introduction, almost all of which
focuses on specific graph sequences, since determining the local antimagic chromatic number for a given graph is typically difficult. The particular graphs studied in this paper are related to the friendship graphs, studied by Arumugam et al., but are harder to analyse. Consequently this fits naturally into the existing literature. The paper also gives a new family of counterexamples to an erroneous result on joins stated by Arumugam et al. (this result is well-known to be incorrect, with two other families of counterexamples previously found). The paper is generally well
written (although there are a few grammatical errors), and proofs are clear, careful and correct. I therefore recommend publication after minor revisions, mostly to the introduction. I detail suggested changes below.
- Throughout, the "\mapsto" arrow is used between domain and codomain. This should be a simple arrow (use \rightarrow).
- line 10 "if for any two adjacent vertices u and v in G satisfying with" -> "if any two adjacent vertices u and v in G satisfy" (and again on line 40)
- line 12 "with assigning" -> "by assigning"
- line 40/41: delete the definition of the notation w(u), E(u), since these were already defined in line 29/30
- line 44: Mention here that [8] shows that every connected graph with at least three vertices is local antimagic.
- line 50: "where n is the common vertex of n triangles of F_n" -> "which consists of n triangles with a common vertex".
- line 60: mention that another counterexample was independently found by Shaebani, and add a reference to the following paper.
- S. Shaebani, On local antimagic chromatic number of graphs, J. Algebraic Systems 7, no. 2, (2020), 245-256.
- line 80/81: This should just be a normal citation, so say "to a theorem of [2]".
- line 83: Instead of saying antimagic graphs are local antimagic, again you should quote the stronger fact that any connected graph with at least 3 vertices is local antimagic as shown in [8].
- Throughout the proofs, words such as "and", "where", etc., do not look right - use e.g. "\mathrm{if } i \mathrm{ is even}" to fix this.
- References [4,5,6] have the authors' names wrong (family name has not been correctly identified). These should all be by "G.-C. Lau, W.-C. Shin and H.-K. Ng."
- Reference [5] should say "Discuss. Math. Graph Theory" (the first word is missing), and the year should be 2021 (the year of publication).
- References [7] and [8] should both be cited in the text at some point (the other references already are). I have already suggested places to cite [8], e.g. line 44. [7] was used in [2] to determine local antimagic chromatic numbers of complete bipartite graphs, so this could be mentioned and a citation added around line 55.
- Reference [7]: the author's name is "Hagedorn" (the last letter is missing).
- There is some inconsistancy between listing authors in the references as e.g. "Hartsfield, N." or "J. Bensmail" - either style is fine but do the same for all references.
Reviewer 2 Report
The results obtained are very interesting and new. Generally, the proofs are presented clearly and easy to follow. However, there are a few typos spotted. In the attached pdf file, more comments are given for the authors to improve the manuscript accordingly. It is also suggested that the authors give a more interesting motivation to this research, and to give a few interesting open problems for future research. This will make the paper more interesting to the readers.

Reviewer 3 Report
In this paper, the authors give the local antimagic chromatic numbers of three classes of join graphs. The proof is a little involved. However, there are some writing errors. Therefore, I recommend this paper be accepted for publication in Journal of Mathematical and Computational Applications with some revision. My detailed comments are attached.
